# The Selection of Efficient Antiscalant for RO Facility, Control of Its Quality and Evaluation of the Economical Efficiency of Its Application

**DOI:** 10.3390/membranes13010085

**Published:** 2023-01-09

**Authors:** Dmitry Spitsov, Htet Zaw Aung, Alexei Pervov

**Affiliations:** Department of Water Supply and Wastewater Treatment, Moscow State University of Civil Engineering, 26, Yaroslaskoye Highway, 129337 Moscow, Russia

**Keywords:** reverse osmosis, calcium carbonate, scaling mechanism, phosphonic-based antiscalants, inhibitors, crystallization mechanism, membrane cleanings, concentrate disposal

## Abstract

Adsorption of polymeric inhibitor molecules to calcium carbonate crystal surface was investigated. Inhibiting efficiencies of phosphonic acid-based antiscalants are dependent on the amount of adsorbed material on the growing crystal surface. A strong antiscalant even at a small dose provides the necessary rate of adsorption. Comparison of two phosphonic-based antiscalants was made both in laboratory and industrial conditions. A distinguishing feature of the strong antiscalant is the presence of aminotris (metylene-diphosphonic acid) ATMP. Experimental dependencies of antiscalant adsorption rates on the antiscalant dosage values were determined. Emphasis is given to the use of nanofiltration membranes that possess lower scaling propensities. Modernization is presented to reduce operational costs due to antiscalant and nanofiltration membranes. The main conclusion is that control of scaling should be implemented together with the use of nanofiltration membranes.

## 1. Introduction

Application of antiscalants to control sparingly soluble salts that deposit in reverse osmosis membrane units has become one of the most important and significant issues in reverse osmosis practice [1,2,3,4]. As was already discussed [5,6,7], proposed antiscalants with new unknown brands require evaluation of their efficiencies. but antiscalant testing results provided by different researchers often do not provide real conditions for crystal nucleation that exist in membrane industrial devices. This often does not enable us to efficiently apply new antiscalants and evaluate their behavior [8,9,10]. Often laboratory testing provides inconsistent results that cannot be transferred into practice [1,8]. In previously published articles several types of phosphonic and acrylic inhibitors have been tested to compare their effectiveness. Comparison of adsorption rates and adsorption capacities was conducted. Now we want to demonstrate the benefits of using reduced doses.

Phosphonates or phosphonic acid-based antiscalants are still recognized as leading products in the reverse osmosis desalination market [5,10]. This leadership is still maintained despite the diversity of their inhibiting abilities. A lot of research was undertaken to develop and introduce into practice “green” antiscalants and a variety of different classes of antiscalants: acrylic acid antiscalants, polyaspartate-based antiscalants, organophosphates and others [9,10,11,12]. This is attributed to the reliability and efficiency of phosphonic-based antiscalants over other classes of inhibitors. Yet belonging to the class of phosphonates does not guarantee high efficiency. Operational experience has demonstrated that different antiscalants even among phosphonates demonstrate different efficiencies that results in certain operational costs, losses and damages. This article aims to demonstrate the differences in antiscaling efficiencies of different phosphonates. As will be discussed, the difference in phosphonates’ behavior is mainly attributed to the amount of aminotris (methylene-phosphonic acid)—ATMP—that is formed during antiscalant synthesis. Figure 1 demonstrates spectrum results of nuclei-magnetic resonance analysis that show the ratio of ATMP and nitrilo-trimethyl-phosphonic acid (NTMP). The “Jurby-Soft” inhibitor consists entirely of NTMP and does not contain ATMP. This antiscalant demonstrated the lowest efficiency. Different phosphonic-based antiscalants demonstrate high efficiencies, but among them “Aminat-K” demonstrates maximum value of NTMP/ATMP ratio (equal to 5:1), corresponding to its maximal efficiency [1,12]. The present article aims to provide experimental proof of this superiority and to demonstrate how antiscalant efficiency ensures reduction of operational costs. These results are very useful for RO management when the most efficient product should be selected and purchased. This method is also very helpful when the quality of the supplied reagent should be checked and the input quality control of the product should be arranged.

At the present time, interest in using reverse osmosis in drinking water supply is growing. Many new suppliers of membrane products and service chemicals are constantly appearing on the market. The tender principle of procurement leads to the situation that a low quality and inefficient product is supplied. The problem is the correct formulation of the antiscalant composition requirements. This is due to an incorrectly formulated requirement to supply a mixture of sodium salts and phosphonic acids, which is not enough to solve the problem. The initial choice of the antiscalant should be based on the results of the analysis, but later, when large amounts of antiscalants are purchased, we should analyze antiscalant samples’ nuclear magnetic resonance spectrums to ensure that the required product is used. In this article, the authors make an attempt to demonstrate a comparison of antiscalant efficiency with an account of the reduction of operational costs due to the dosing of high-quality product in the feed water.

## 2. Experimental Part: Materials and Equipment

The goal of the experiments was to demonstrate the adsorption characteristics of different antiscalants and to connect adsorption ability and antiscaling efficiency of different products with the purpose of developing guidelines to reduce operational costs. The experimental program included:Determination of calcium carbonate crystallization rates depending on antiscalant dose and membrane type.Evaluation of antiscalant adsorption rates for different doses.Dependencies of adsorption rates on the antiscalant type and its dose.

The laboratory test unit flow diagram is shown in Figure 1. The evaluation of scaling rates was conducted in accordance with the test procedure developed previously and described in a number of publications [1,12]. The test procedure was based on operation of the test unit in recirculation mode whereby concentrate of membrane module was returned back to the feed water tank and permeate was constantly withdrawn from the system and collected in a separate tank [1]. Underground water was used in the experiments. Samples were taken from water intake wells at reverse osmosis water treatment plants in the Moscow area. The TDS value of the ground water was 700 ppm, calcium concentration was 5.6 milliequivalents per liter; alkalinity was 120 ppm; chloride concentration was 210 ppm; sulphate concentration was 288 ppm; pH was 7.2. Water composition of samples taken from three different wells is presented in Table 1. The test procedure included circulation of the feed water in the membrane unit and evaluation of scaling rates using the method of mass balance.

The amount of calcium carbonate deposited on the membrane surface was determined by mass balance considerations as a difference between the amount of calcium in the feed water tank in the beginning of experiment and the amount of calcium carbonate at the moment of the experiment. Deposition rates of calcium carbonate were defined as the derivative of the function of the amount of deposited calcium over time.

Feed water (ground water in the Moscow region) was added to feed water tank 1 and was then pumped by pump 2 to membrane module 3. Membrane modules model 1812 70 NE with nanofiltration membranes and 1812 BLN with low pressure reverse osmosis membranes were used. The Procon rotary pump was supplied by Procon Products, Smyrna, TN, USA and produced 180–200 L per hour at a pressure of 16 bar. The experiments were carried out using serial membrane elements of the 1812 standard model produced by Toray Advanced Materials Korea Inc. (the manufacturer of CSM Membrane Technologies, Korea, Seoul company CSM) with reverse osmosis membranes of the BLN model (selectivity for salts of 95–96%) and nanofiltration elements of the model with membranes of the 70 NE type with a selectivity of 70%. The area of the membranes in the apparatus model 1812 was 0.5 square meters.

Concentrate samples were taken from tank 1. Calcium, chloride, sulphate and bicarbonate ions concentrations as well as pH and TDS values were determined in the samples. The test procedure and calculation techniques to evaluate scaling rates in the presence of antiscalants and without their addition has been discussed in a number of publications [1,12]. Figure 2a shows dependencies of calcium concentrations in tank 1 on the coefficient K value (that is determined as a ratio of the initial volume in tank 1 to the concentrate volume at the moment of the experiment). The amount of calcium carbonate deposited in the membrane channel was calculated as the difference between calcium amount in tank 1 at the beginning and at the end of each test run (Figure 2b). Figure 2c shows dependencies of calcium carbonate deposited in the membrane module as a function of time. Scaling rates were determined as tangent values of calcium carbonate amount M (in milliequivalents) versus time T function and were expressed in milliequivalents per hour (Figure 2d). Scaling rates were determined for antiscalant doses 2, 5 and 10 ppm for both antiscalants (Figure 2d). During experimental test runs, antiscalant adsorption rates were also evaluated. Figure 3 shows the results of determination of antiscalant adsorption rates to crystal surface. Adsorption rates were determined in accordance with the previously developed and described procedure [1,2]. The rejection characteristics of membranes also significantly influence scaling propensities of membrane modules [13,14,15,16]. Figure 3 demonstrates results of evaluation of calcium carbonate scaling rates in membrane 1812 elements tailored with nanofiltration 70NE membranes. As is shown on Figure 3b, scaling rates in nanofiltration membrane elements are at least 4–5 times lower than in reverse osmosis elements.

During the conducted test runs, concentration values of antiscalants (concentrations of phosphate ions) were determined (Figure 4a). Dependencies of antiscalant adsorption rate values on K values are shown on Figure 2b. The higher the antiscalant dose, the higher the adsorption rate. A relationship between adsorption rates and antiscalant doses is shown in Figure 5. These relationships are built for the K value of 5. These relationships demonstrate the comparison of properties of different antiscalants.

## 3. Experimental Results

### 3.1. Effect of the Antiscalant Dose on Scaling Rates

Figure 2 shows the main steps in determining calcium carbonate scaling rates in the presence of antiscalants and without antiscalant addition. The amount of calcium carbonate was determined by a mass balance as a difference between the amount of calcium in tank 1 at the beginning of the experiment and calcium amount in tank 1 at the moment of the experiment [6]. Figure 2a demonstrates dependencies of deposited calcium carbonate amounts on coefficient K values. Calcium carbonate deposition rates were determined in conformity with the method described in [6] as tangents of the function of calcium carbonate amount versus time (Figure 2b). Results of scaling rate determination are demonstrated in Figure 2c as dependencies of calcium carbonate scaling rate values (expressed as milliequivalents of calcium carbonate per hour) on coefficient K value. Figure 2c demonstrates coefficient K values that correspond to beginning of scaling in the RO module under different conditions in the presence of antiscalants and without antiscalant addition. Different K values correspond to different scale formation (nucleation) conditions in the “dead” area [2].

### 3.2. Effect of Membrane Type on Scaling Rates

Aminat-K demonstrates lower scaling rates than Jurbi-Soft at different doses. The membrane type also influences scaling rate. As can be seen in Figure 3, scaling rates in the nanofiltration membrane element are substantially lower than in the membrane element with reverse osmosis membranes under all the same experimental conditions: water composition, pressure, recovery, antiscalant dose. Reduced scaling rates in nanofiltration membrane modules are attributed to low rejection values and lower supersaturation conditions in the “dead” areas in membrane channels which creates the conditions for the start of crystallization [1,12]. The use of low rejection membranes in drinking water production projects can be considered along with antiscalant addition as a measure to control scaling [2].

### 3.3. Evaluation of Antiscalant Adsorption Rates and Their Influence on Scaling Rates

Figure 4 and Figure 5 demonstrate results of Aminat-K and Jurby-Soft adsorption rate measurements during calcium carbonate deposition on membrane surface. Aminat-K exhibits higher adsorption abilities than Jurby-Soft at the same doses. Stronger Aminat-K provides higher adsorption rate and higher calcium carbonate scaling rate reduction at 10 ppm dose when coefficient K equals 1.5, but Jurby-Soft cannot provide such an efficiency value and such an adsorption rate with a 10 ppm dose. Figure 5 shows that Aminat-K is a strong antiscalant and even at a small dose of 2 ppm provides higher adsorption rates than Jurby-Soft. This also explains why Aminat-K shows the same results in the concentration range from 2 to 7 ppm while Jurby-Soft gives different results under the same conditions.

### 3.4. Selecting Antiscalant and Membrane Type to Reduce Operational Costs

Antiscalant suppliers are usually limited by general recommendations to add 1 to 10 milligrams of antiscalant per one liter of RO feed water. More detailed recommendations can be provided in accordance with the calculations that suppliers have developed for their product for a variety of application conditions. Thus, the dosing of antiscalant changes depending on supersaturation ratio reached in the membrane module depending on feed water composition and recovery values [1]. Calculations often give inconsistent and controversial results as they depend on the experimental conditions under which the crystal formation and scale deposition were conducted [1,2]. In our experiments we evaluate the scaling rates using the ground water from the source, industrial membrane module and pressure and recovery conditions relevant to the industrial operation conditions of the RO plant. This enables us to provide a rapid quantitative evaluation of scale deposition rate in the presence of different amounts of antiscalant and to find an answer: which makes it possible to reduce the dose without compromising the effectiveness of scaling control.

Application of antiscalants provides reduction of scaling and higher values in the period between the cleanings. Figures demonstrate results of scaling rate evaluations as a function of coefficient of initial volume reduction K for the cases of different antiscalants and different dose applications. Figure 6a shows the dependence of accumulated calcium carbonate amount on time of operation that was obtained previously as a result of investigations [2,16] using 1812 elements tailored with low pressure RO and NF membranes. These relationships are used for prognosis of product flow and rejection. Figure 6b shows the recommended time for membrane cleaning for the presented case study; time periods T1, T2 and T3 correspond to the exact amount of calcium carbonate (600 milliequivalents) deposited on 1 square meter of membrane.

Table 1 shows the main ground water compositions met in water intake in the Moscow region. Hardness can vary between 5 and 7 milliequivalents and alkalinity between 4 and 6 milliequivalents per liter. To predict antiscaling behaviour of Aminat-K antiscalant in a variety of conditions, authors have developed software to help RO operators to select recoveries and cleaning schedules. For three different feed water compositions presented in Table 1, results of calcium carbonate scaling rate evaluation are presented (Figure 7a) as a function of coefficient K value and recommended time of operation between cleanings is calculated (Figure 7b). For cases when a different new antiscalant (Jurby-Soft) is used, Figure 7 shows results of scaling rates and time period between cleanings as a comparison.

In cases when another antiscalant is used which demonstrates lower efficiency, the time period between cleanings decreases. Very often. applied cleanings are economically unreasonable. If we extend the period between cleanings, the amount of accumulated calcium carbonate grows and the efficiency of membrane cleaning decreases [2,16]. Thus, a certain amount of scale is left after routine cleanings and constantly grows.

## 4. Industrial Application of the Results

Antiscaling efficiency of the dosed antiscalant provides reduction in scaling rates and longer operational period between cleanings. Figure 6 shows the results of scaling rates determination as a dependence on K value The plots shown in Figure 6a are used for prognostic techniques to recommend operational time for predicted product flux and rejection decrease, based on results of experimental evaluation of scaling rates [1,2]. Figure 6b recommends an operational time value of 300 h for the case of ground water composition and K value of 4. For cases when different antiscalants are used with higher or lower scaling rates and retardation efficiencies, operational time required to reach cleaning should be either shortened or expanded. In our case, replacing of Jurby-Soft antiscalant by Aminat-K provides nearly 50% reduction of scaling rates and therefore ensures safe operation of the reverse osmosis facility within 600 h instead of 300. As discussed earlier (Figure 2a), Aminat-K provides efficient scaling rate reduction even using reduced dose values (1.5–2 ppm). Thus, total annual operational cost savings due to application of Aminat-K include reduced annual consumption of cleaning reagents and reduced consumption of antiscalant. Table 2 describes the calculation steps for evaluation of annual operational costs for the reverse osmosis plant operated in the Moscow region with capacity of 130 cubic meter per hour. The plant consists of two parallel lines, each including nine pressure vessels with six membrane 8040 elements (Figure 7). Figure 8 and Figure 9 describe the membrane vessels’ array and the graph shows the increase of K while water is moved from the first element to the last, or “tail” element, as well as scaling rate values. This helps us to evaluate scaling rates in each membrane element and total scaling rate and total amount of calcium carbonate scale accumulated in the array (Figure 6). The amount of required cleaning reagent (usually citric acid and EDTA) and cleaning schedules are taken according to results of cleanings. Figure 10 shows results of routine cleanings applied every 300 h of continuous operation (on average once a month) with citric acid. Assuming volume of cleaning tank as 3 cubic meters, it can be calculated that an amount of 1500 equivalents can be removed using three cleaning series. Citric acid concentration was 3% (Figure 11).

Another expense item is concentrate disposal. Concentrate produced by a reverse osmosis plant used for drinking water supply is forwarded to a sanitation sewer. For cases when ground water in Moscow region is treated by RO, the K value is usually 3–4, which means that concentrate flow value is 1/3 to 1/4 of the feed water flow. Thus, the water user water tax includes not only the costs for water treatment but costs for additional wastewater discharge. To reduce concentrate flow, an additional stage of nanofiltration membranes is used [1,16]. Concentrate enters nanofiltration membrane modules where it is separated into the product stream and concentrate stream that equals 1/10–1/20 of the feed water flow. Nanofiltration membrane produce water is added to the feed ground water and this increases recovery and the amount of water taken from the well and pretreated. Figure 10 shows the flow diagram of membrane modules array and the graph shows scaling rate increase due to K value growth. When purchasing large amounts of antiscalant to avoid fakes, it seems reasonable to apply analysis to the samples of the product to be sure that the product has required MIDF concentration.

## 5. Conclusions

Antiscaling efficiency of the inhibitor depends on its ability to adsorb on the surface of growing crystals. The higher adsorption rate, the better the antiscaling behavior of the inhibitor.To efficiently control scaling and to reduce operational costs of the RO unit it is recommended to tailor the membrane plant with nanofiltration membranes. The joint use of low rejection membranes and efficient antiscalant provides substantial decrease in scaling rates in membrane modules and reduces operational costs.High adsorption abilities of phosphonic-based antiscalant enables us to reduce the antiscalant dose in the feed water without compromising effectiveness of scale control and thus reduce reagent consumption and operational costs.High efficiency of phosphonic antiscalants is attributed to the content of aminotris (methylene=phosphonic acid) in the product. Application of Nuclear Magnetic Resonance method helps to identify the presence of ATMP and to avoid buying low-quality products.

## Figures and Tables

**Figure 1 membranes-13-00085-f001:**
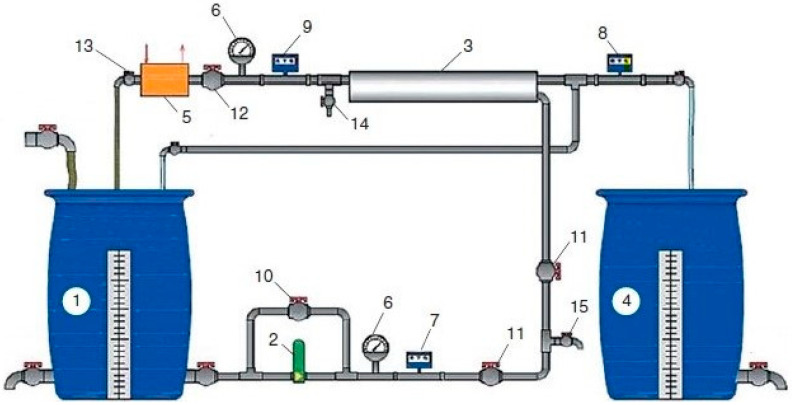
Laboratory test unit flow diagram: 1—source water tank; 2—pump; 3—membrane element in the pressure vessel; 4—filtrate tank; 5—heat exchanger; 6—manometer; 7–9—flow meters; 10—bypass valve; 11—valve for adjusting the flow of source water; 12—valve for adjusting the working pressure and concentrate flow; 13—valve for adjusting the flow of cooling water; 14, 15—samplers.

**Figure 2 membranes-13-00085-f002:**
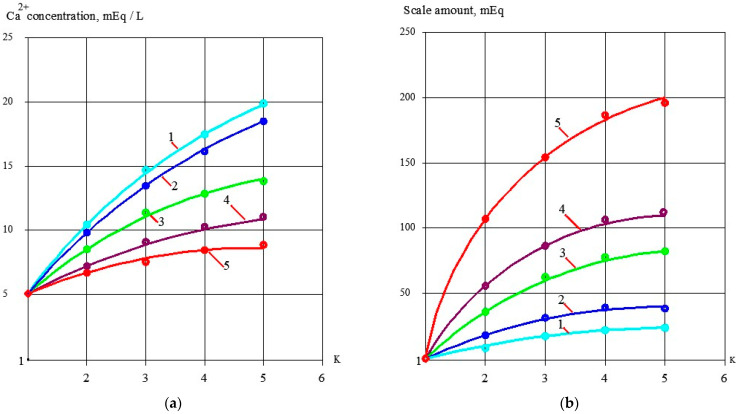
Evaluation of calcium carbonate scaling rates in membrane elements with reverse osmosis low pressure membranes: dependences of concentrations of calcium in concentrate volume on K values (**a**); dependencies of accumulated calcium carbonate amount on K (**b**); dependencies of calcium carbonate amount on time of experiment T (**c**);dependencies of calcium carbonate scaling rates on K (**d**); 1—Aminat-K, dose 10 ppm; 2—Aminat-K, dose 5 ppm; 3—Jurby-Soft, dose 10 ppm; 4—Jurby-Soft, dose 5 ppm; 5—without antiscalant.

**Figure 3 membranes-13-00085-f003:**
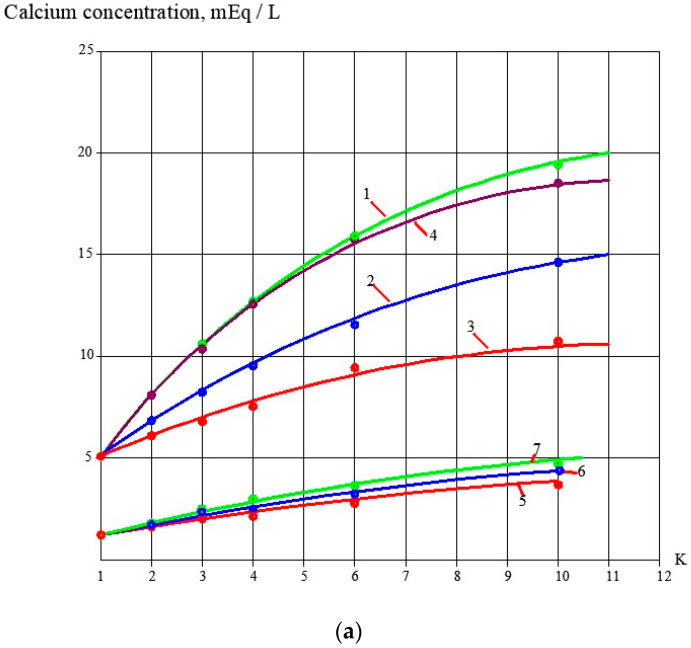
Results of evaluation of calcium carbonate scaling rates in membrane elements with nanofiltration membranes: dependencies of concentration of calcium in concentrate volume on K values (**a**); dependencies of calcium carbonate scaling rates on K values (**b**); 1—Aminat-K, 5 ppm; 2—Jurby-Soft, 5 ppm; 3—without antiscalant addition; 4—Aminat-K, 2 ppm; 5—product flow after addition of 5 ppm of Aminat-K; 6—product flow after addition of 2 ppm of Aminat-K; 7—product flow without antiscalant addition.

**Figure 4 membranes-13-00085-f004:**
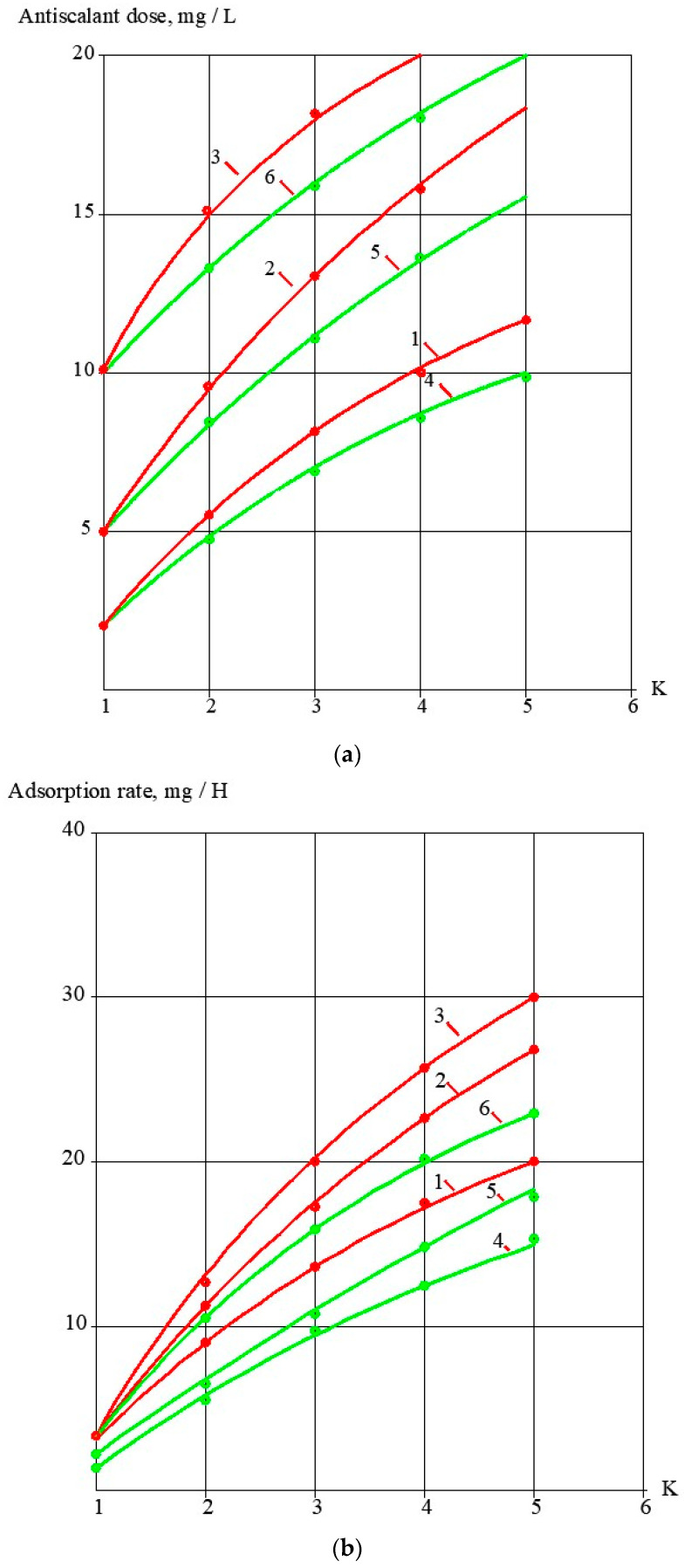
Dependencies of Aminat-K and Jurby-Soft doses in RO concentrate on K values (**a**) and dependencies of adsorption rate values on K (**b**): 1—Aminat -K, dose 2 ppm; 2—Aminat-K, dose 2 ppm; 3—Aminat-K, dose 10 ppm; 4—Jurby-Soft, 2 ppm; 5—Jurby-Soft, 5 ppm; 6—Jurby-Soft, 10 ppm.

**Figure 5 membranes-13-00085-f005:**
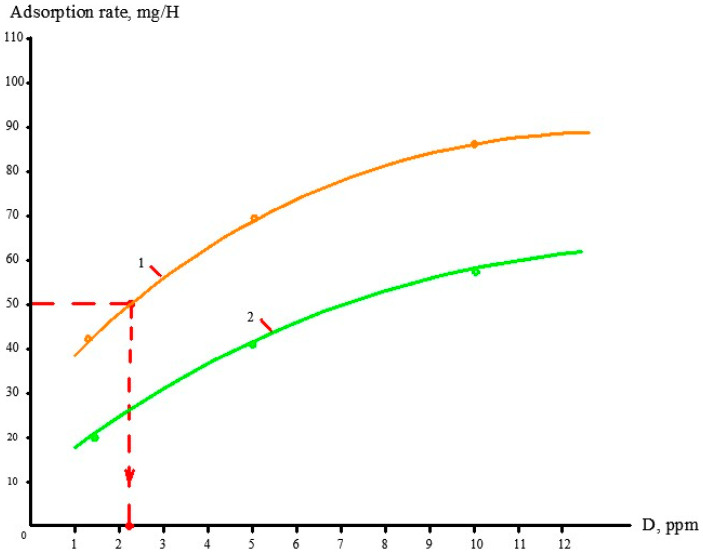
Dependencies of antiscalant adsorption rates on the antiscalant dose values: (1)—Aminat-K; (2)—Jurby-Soft.

**Figure 6 membranes-13-00085-f006:**
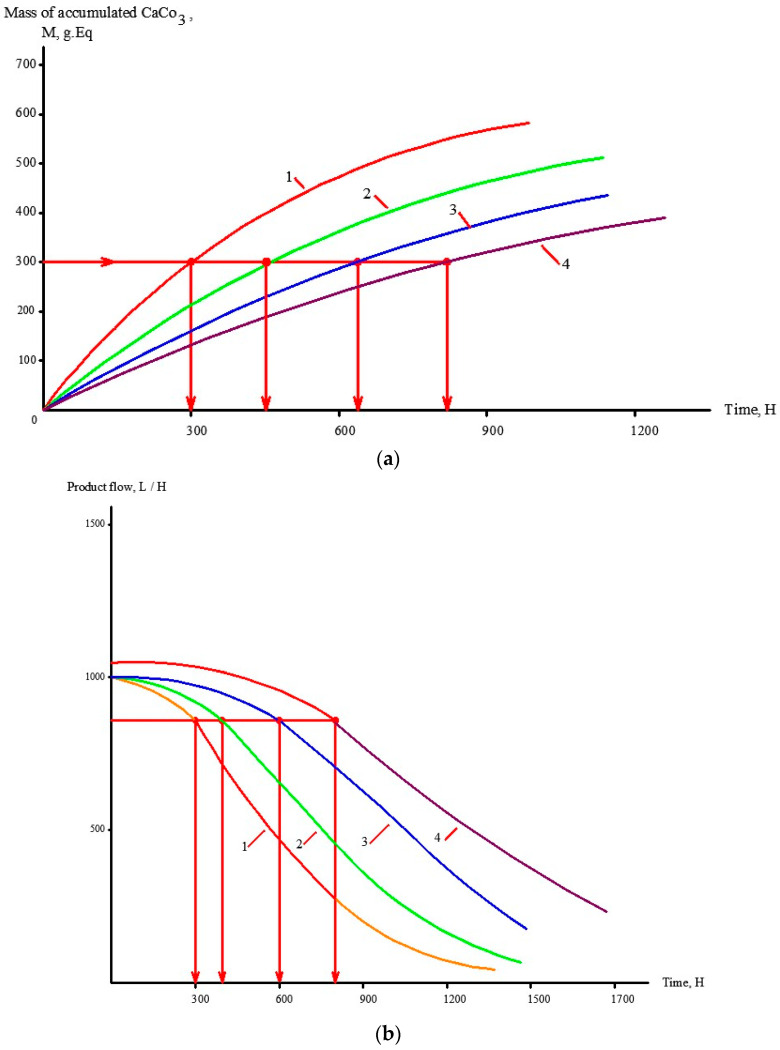
Prediction of membrane product flow decrease with calcium carbonate accumulation over time and recommended time between membrane cleanings: (**a**)—dependencies of accumulated calcium carbonate amount on time; (**b**) dependencies of product flow on time; 1—Jurby-Soft, 5 ppm; 2—Jurby-Soft, 10 ppm; 3—Aminat-K, 5 ppm; 4—Aminat-K, 10 ppm.

**Figure 7 membranes-13-00085-f007:**
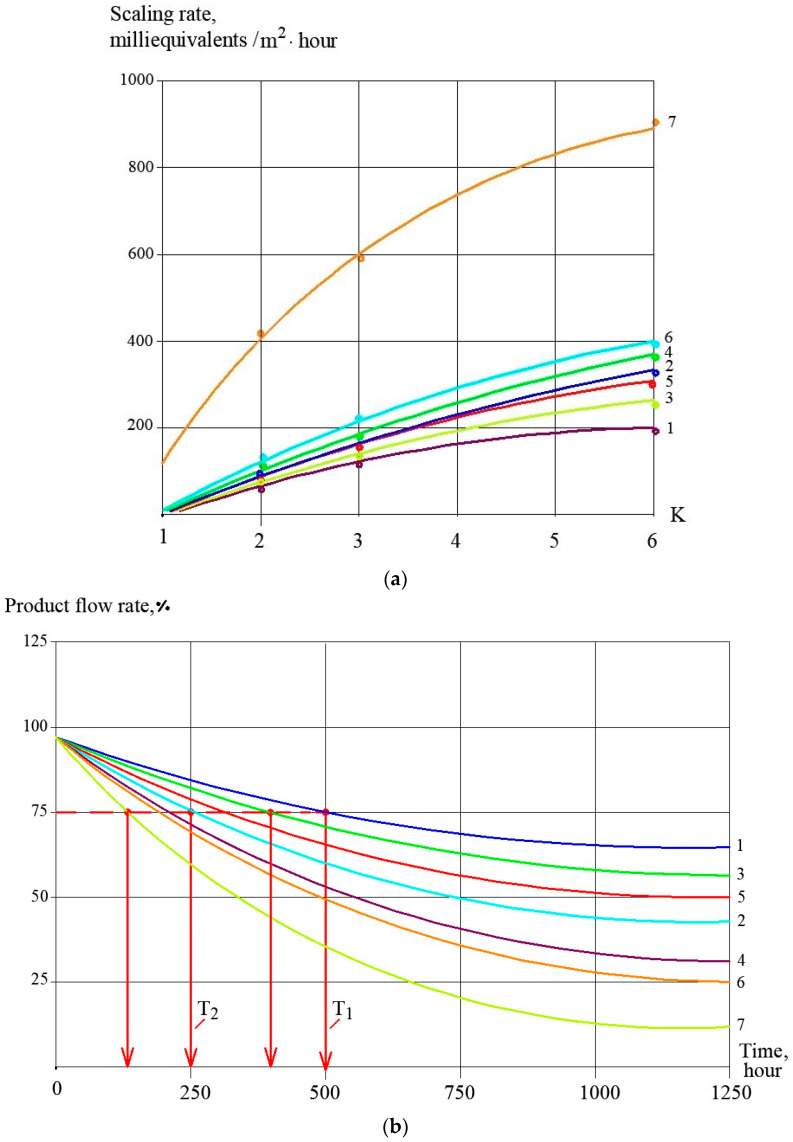
Prediction of membrane product flow decrease with calcium carbonate accumulation over time and recommended time between membrane cleanings: (**a**)—dependencies of accumulated calcium carbonate amount on time; (**b**) dependencies of product flow on time; 1—water composition 1, Aminat K 5; 2—water composition 1, Jurby Soft 5 ppm; 3—water composition 2, Aminat K 5; 4—water composition 2, Jurby Soft 5; 5—water composition 3, Aminat K 5; 6—water composition 3, Jurby Soft 5 ppm; 7—water composition 1, without antiscalant addition (Table 1).

**Figure 8 membranes-13-00085-f008:**
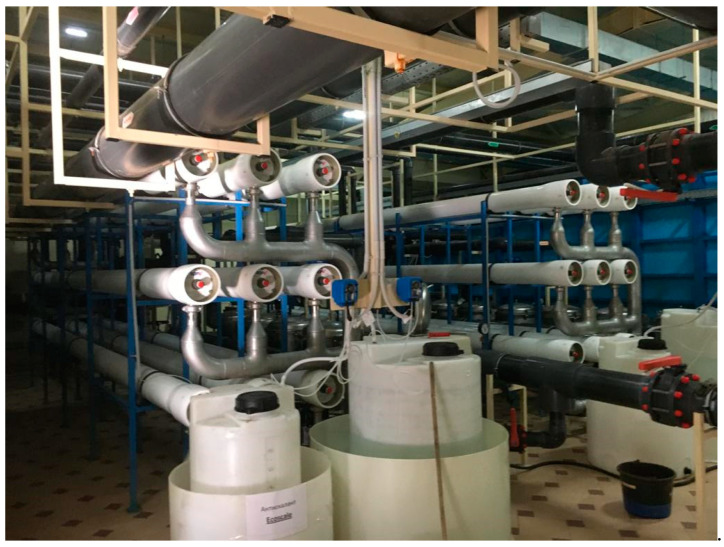
A 120 cubic meter per hour reverse osmosis membrane plant for production of drinking quality water from the ground water well intake in the Moscow region.

**Figure 9 membranes-13-00085-f009:**
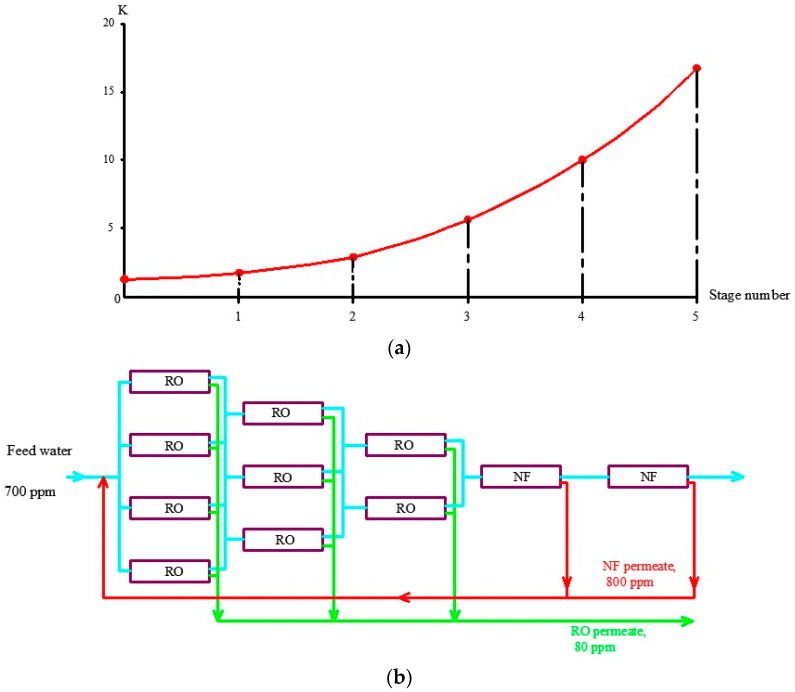
Evaluation of K values in membrane elements of the RO plant array: (**a**) dependencies of K values on the membrane stage number; (**b**) pressure vessels array in membrane plant.

**Figure 10 membranes-13-00085-f010:**
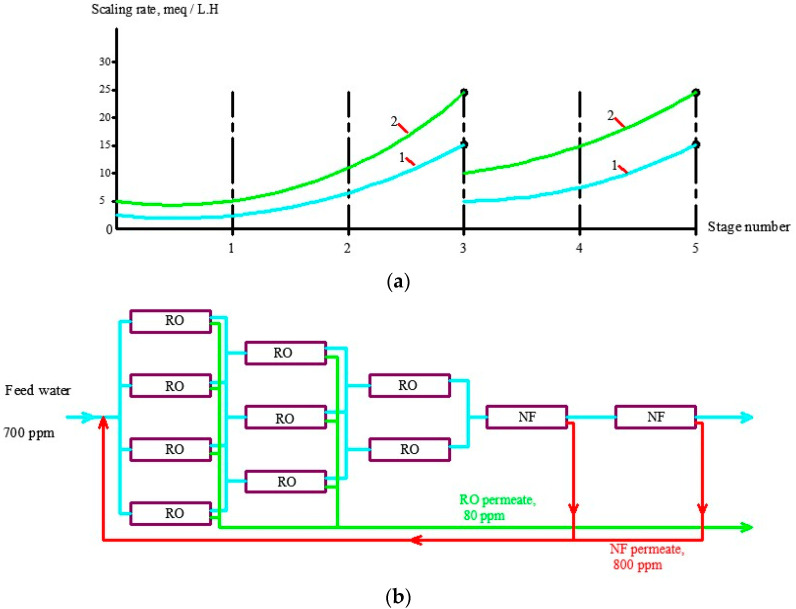
Evaluation of calcium carbonate scaling rates in membrane elements of the RO plant array: (**a**) dependencies of scaling rates on the membrane stage number; (**b**) pressure vessels array in membrane plant; 1—Jurby-Soft, 5 ppm; 2—Aminat-K, 2 ppm.

**Figure 11 membranes-13-00085-f011:**
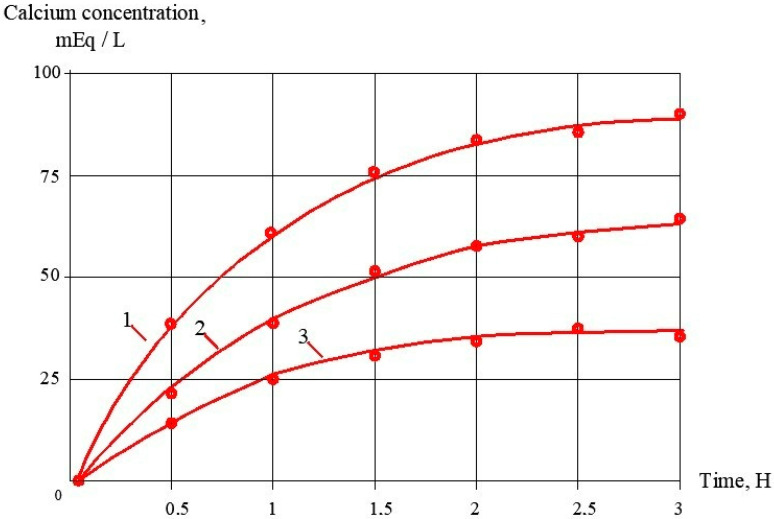
Dependencies of calcium ionic concentration in citric acid cleaning solution on time: 1—first cleaning; 2—second cleaning; 3—third cleaning.

**Table 1 membranes-13-00085-t001:** Ground water composition in different intakes (Moscow region).

	Component	Intake 1	Intake 2	Intake 3
1	Ca^2+^	4.4	3.9	3.5
2	Mg^2+^	2.3	1.4	2.5
3	SO_4_^2−^	0.63	0.2	0.7
4	Cl^−^	0.29	0.3	0.15
5	HCO_3_^−^	6.5	5.8	6
6	PH	7.2	7.3	7.2
7	TDS	590	510	480

**Table 2 membranes-13-00085-t002:** Evaluation of operational costs: economical comparison of antiscalant efficiencies and recovery increase due to application of nanofiltration.

	Characteristics	Present Technique (Antiscalant Dose 6 ppm)	Recommended Technique (Antiscalant Dose 4/2 ppm)
1	Feed flow, (m^3^/H)	130/910,000	130/910,000
2	Product flow, Designed, (m^3^/H)/ Real, (m^3^/year)	100 700,000/ 520,000	100 700,000/ 630,000
3	Concentrate flow, $m^{3}/H$	30	30/1.5 (After recovery increase using NF)
4	Pressure, Bar	14–16.5	14–15
5	Annual power consumption, KW	112,000	105,000
6	Power costs, USD/year	8000	7500
7	Antiscalant type, and dose, ppm	Jurbysoft, 6 ppm	Aminat-K, 4/2
8	Annual antiscalant consumption, kg	5460	3640/1820
9	Antiscalant costs, USD per ton	2857	3571
10	Annual antiscalant cost, USD	15,600	13,000/6500
11	Number of cleanings per year	8	4
12	Reagent consumption; Caustic cost, USD/liter, Acid cost, USD/liter	6000/100 4800/120	3000/100 2400/120
13	Annual cleaning costs; caustic, USD acid, USD	8571 8229	4286 4114
14	Membranes; model/number/cost, USD	Lewabrane RO B 370 HF/108	500
15	Annual costs for membrane replacement, USD	9000	9000
16	Annual operational costs, USD	49,400	31,400
17	Water cost price, USD/ $m^{3}$	0.097	0.05
18	Membrane elements to reduce concentrate flow: model/number/cost, USD	8040 70 NE/24/750	
19	Additional annual costs for membrane replacement, USD	-	3000
20	Concentrate discharge flow, $m^{3}$/hour/tax, USD/$m^{3}$	-	0.33
21	Annual payment for concentrate discharge, USD	70,000	3500

## Data Availability

The data presented in this study are available on request from the corresponding author.

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
