# Peer review of "The Selection of Efficient Antiscalant for RO Facility, Control of Its Quality and Evaluation of the Economical Efficiency of Its Application"

_membranes, 2023, doi:10.3390/membranes13010085_

Round 1

Reviewer 1 Report

The authors studied the effect of phosphonic based antiscalant content (ATMP amount) and dose on calcium carbonate crystallization both in lab scale and in an industrial system. Their goal was to explain why high efficiencies are attained at high NTMP/ATMP ratio and to demonstrate how antiscalant efficiency reduces operational costs. The study looks interesting but needs an extensive revision before being accepted, taking those points on mind:

1.          The feed water used is only one type (ground water from an industrial plant, with relatively low TDS of 700ppm). While this gives a reliable result on the performance of antiscalants for that specific plant, the results may not be generalized for other plants where water feed quality and operating conditions are different. To tackle this, the authors may utilize the lab scale setup to test different feed water qualities (i.e. TDS of 700, 2000,…). In addition, feed water pH is an important parameter to vary since it has a huge effect on calcium carbonate crystallization.

2.       The results section needs to be reorganized and restructured with the introduction of subsections discussing the findings on a logical order (i.e.  1. Effect of membrane type on scaling rates, 2. Effect of Antiscalant dose and Adsorption rate on scaling rate, 3. How does 1&2 translates to operational costs minimization. In addition, the figures on this section could be better represented by fixing the axis titles in proper locations, deleting grid lines, ….

Other comments to improve the quality of the paper:

1.       Grammatical mistakes should be taken care of throughout the whole manuscript, and specifically in lines 29-32, 35-38, 62, and 90.

2.       In line 85, an explanation of the scaling rate evaluation method used needs to be added then supported by the Refs.

3.       Add the concentrations of chloride and sulphate ions in lines 92 and 93.

4.       A justification of the doses of antiscalants used (2, 5, 10 ppm) in line 115 needs to be discussed? Are those the optimum doses giving the highest efficiency? Are those recommended by the suppliers based on the water feed quality?

5.       The in text figure caption in lines 141, 145, and 146 need to be corrected.

6.       Add figure number in lines 186, and 191.

Author Response

Dear reviewer,

We would like to thank you for your comments, which helped to improve the quality of the article. Please find a detailed description of the comments and their consideration in the article below.

Comment 1: The feed water used is only one type (ground water from an industrial plant with relatively low TDS of 700 ppm). While this gives a reliable result on the performance of antiscalants for that specific plant, the results may not be generalized for other plants where water feed quality and operation conditions are different. To tackle this, the authors may utilize the lab scale setup to test different feed water qualities (i.e. TDS of 700, 2000,...) In addition, the feed water pH is an important parameter to vary since it has a huge effect on calcium carbonate crystallization.

Reply to the comment 1:

Yes. The authors are grateful to the reviewer for opening this discussion. Exactly, the introduction of the commercial antiscalant into the market requires a thorough examination and research. As authors work for a long time in Russian market, we support the main popular antiscalant Aminat-K with a long impressive list of results and advantages over a number of different chemicals in Russian market. Similar to our foreign colleagues, we have conducted a long research to develop a software to recommend application of Aminat-K in a variety of conditions where scaling rates were evaluated depending on water origin, TDS, calcium hardness, pH, alkalinity as well as on membrane type, rejection, water pressure, recovery etc. As a result, the software was developed to encourage RO plant operators to select operational mode and cleaning schedules. The goal of the present article was to convince to use the Aminat-K and to refrain from purchasing antiscalants using the tender where only in these cases suppliers are mainly selling the product according to the ordered composition. We tried to demonstrate that inhibition ability of the reagent depends on it's absorption qualities. And this provides better solution and saving operational costs. But we decided, basing on our data, to demonstrate the Aminat results to show how the composition influences scaling and schedule.

We added Figure 7 and the following text:

"Table 1 shows main ground water compositions met in water intakes in Moscow region. Hardness can vary between 5 and 7 milliequivalents and alkalinity between 4 and 6 milliequivalents per liter. To predict antiscaling behaviour of Aminat-K antiscalant in a variety of conditions, authors have developed of software to help RO operators to select recoveries and cleaning schedules. For three different feed water compositions presented in the Table 1 results of calcium carbonate scaling rate evaluation are presented (Figure 7,a) as a function of coefficient K value and recommended time of operation between cleanings is calculated (Figure 7,b). For the cases when a different new antiscalant (Jurby-Soft) is used, Figure 7 shows results of scaling rates and time period between cleanings comparison".

Comment 2: The Results section needs to be reorganized and restructured the introduction of subsections, discussing the findings on a logical order. In addition, the figures on this section could be better represented by fixing the axis title in proper locations, deleting grid lines.

Reply to comment 2: Yes, we definitely agree and have implemented this restructuring by introducing new subsections: 3.1 Effect of membrane type on scaling rates; 3.2 Effect of the antiscalant dose on scaling rates; 3.3 Evaluation of antiscalant adsorption rates and their influence on scaling rates. 3.4. Considerations to reduce operational costs selecting antiscalant and membrane types.

  1. Other comments and improvements:

3.1. We made improvements of the text in lines 29-32; 35-38; 62; 90.

3.2. The explanation of the scaling rate evaluation technique is added to the line 85:

The amount of calcium carbonate deposited on membrane surface was determined by a mass balance considerations as a difference between the amount of calcium in feed water tank in the beginning of experiment and the amount of calcium carbonate at the certain moment of experiment. Deposition rates of calcium carbonate was defined as the derivative of the function of the amount of deposited calcium from time [1,12].

3.3. Concentration values of chloride and sulphate ions are added in lines 92 and 93

3.4. "A justification of the doses of antiscalants used in line 115 needs to be discussed. Are those the optimum doses giving the highest efficiency? Are those recommended by suppliers based on the water feed quality? "

Reply to the comments (Comment 3.4):

This comment is very important. Until now a lot of discussions still exist around the antiscalant doses and efficiencies of different groups of antiscalants. Despite the recognition of the certain antiscalant trademarks (such as Genesis, Nalco etc.) at Russian Market, we still experience the principles of tender procurement when the product is proposed according to low est. pricing and approximate chemical composition. Our research group for a long time provides experimental support to the product named "Aminat-K" which is produced and supplied by Traverse Co (Moscow). Thus all doses and efficiencies are evaluated by us and recommended basing on results of experimental research. Similarly, the web pages of the main antiscalant suppliers (for example, "Genesis") recommend doses from 1 to 10 ppm, but the final decision can be made by the user only basing on results of special software calculations or experimental research. Thus, the presented in the article results are aimed to provide comparison of antiscalant efficiencies and to convince the users that selection of antiscalant should be based on scientifically obtained results.

We added the following to the article:

"Antiscalant suppliers are usually limited to general recommendations to add 1 to 10 milligrams of antiscalant per one liter of RO feed water. More detailed recommendations can be provided in accordance with the calculations that suppliers have developed for their product for variety of application conditions. Thus, the dosing of antiscalant changes depending on supersaturation ratio reached in membrane module depending on feed water composition and recovery values [1]. Calculations often give inconsistent and controversial results as they depend on the experimental conditions under which the crystal formation and scale deposition was conducted [1, 2]. In our experiments we evaluate the scaling rates using the ground water from the source, industrial membrane module and pressure and recovery conditions that to industrial operation conditions of the RO plant. This enables us to provide a rapid quantitative evaluation of scale deposition rate in the presence of different amounts of antiscalant. And also to find an answer: in which cases it is possible to reduce the dose without compromising the effectiveness of scaling control."

Also, corrections to lines 141, 145 and 146 are made.

Figure numbers in lines 186 and 191 are added.

Reviewer 2 Report

This study compared the performance of two phosphonic antiscalants in the reverse osmosis (RO) process. The main issue of this manuscript is the insufficient discussion of the experimental results. The authors should add more in-depth discussions about the performance of the antiscalants and the relative mechanism beneath the phenomena. Currently, section 3 and 4 are only descriptions of the results. There are also some minor issues.

1.      From Line 137, Figure 9 was discussed, which is apparently a wrong figure number.

2.      There are 8 keywords in this manuscript, which are too many.

3.      Line 49, the “ATNP” should be “ATMP”.

4.      Figure 1 seems not necessary. Please consider removing it.

Author Response

Dear reviewer,

We would like to thank you for your comments, which helped to improve the quality of the article. Please find a detailed description of the comments and their consideration in the article below.

Comment 1: The authors should add more in-depth discussions about the performance of antiscalants and relative mechanism beneath the phenomena. Currently, section 3 and 4 are only description of results.

Reply to the comment 1:

We added the following:

  1. "Aminat-K demonstrates lower scaling rates than Jurbi-Soft at different doses. The membrane type also influences scaling rate. As it can be seen on Figure 3, scaling rates in nanofiltration membrane element are substantially lower than in membrane element with reverse osmosis membranes under all the same experimental conditions: water composition, pressure, recovery, antiscalant dose. Reduced scaling rates in nanofiltration membrane modules are attributed to low rejection values and lower supersaturation conditions in the "dead" areas in membrane channels which creates the conditions for the start of crystallization [1, 12]. The use of low rejection membranes in drinking water production projects can be considered along with antiscalant addition as a measure to control scaling [2]."
  2. "Figures 4 and 5 demonstrate results of Aminat-K and Jurby-Soft adsorption rate measurements during calcium carbonate deposition on membrane surface. Aminat-K exhibits higher adsorption abilities than Jurby-Soft at the same doses. More strong Aminat-K provides higher adsorption rate and higher calcium carbonate scaling rate reduction at 10 ppm dose when coefficient K equals 1.5. But Jurby-Soft cannot provide such an efficiency value and such adsorption rate with 10 ppm dose. Figure 5 shows that Aminat-K is a strong antiscalant and even at a small dose of 2 ppm provides higher adsorption rates than Jurby-Soft. This also explains why Aminat-K shows the same results in the concentration range from 2 to 7 ppm while Jurby-Soft gives different results under the same conditions."

Also the following corrections are made:

  1. Correction of the figure number is made in discussion, presented in the line 137.
  2. The number of keywords is decreased.
  3. ATMP is corrected.
  4. Figure 1 is removed, we agree that it is not necessary.

Reviewer 3 Report

The manuscript investigated the antiscalants efficiency with account to the reduction of operational costs, which is very useful for the design and operation of RO plant. The manuscript can be published on Membranes after addressing the following two points.

1. The morphology of membrane samples during operation should be given.

2. Please recheck the pressure vessels array in membrane plant in Figure 9 and Figure 10.  Please provide the configuration of brine and permeate flows in Figure 10.   

Author Response

Dear reviewer,

We would like to thank you for your comments, which helped to improve the quality of the article. Please find a detailed description of the comments and their consideration in the article below.

  1. Comment 1:

    The morphology of membrane samples during operation should be given.

    Reply to the comment 1:

    Membrane type used in both reverse osmosis and nanofiltration modules was thin film composite. Membrane material is aromatic polyamide. During experimental program conduction often membrane autopsies are conducted and flat sheet membrane samples are examined to determine morphologies of sedimeted material. Throughout our long research since 1990 we had undertaken a lot of attempts to investigate crystals shapes and size distribution influenced by antiscalant presence [1, 12]. In the present article authors aimed to demonstrate correlation between scaling rates and antiscalant adsorption rate as a main quality that distinguishes “strong” antiscalant. The formation of calcium carbonate on membrane does not disturb antiscalant’s behaviour as we applied cleanings of membrane modules with citric acid (2%) after each test run.

    Comment 2:

    Please recheck the pressure vessels array in membrane plant in Figure 9 and Figure 10. Please provide the configuration of brine and permeate flows in Figure 10.

    Reply to Comment 2.

    All corrections are made with appreciation.

Round 2

Reviewer 1 Report

Authors made significant improvments and they are satisfactory to me.

Reviewer 2 Report

My concerns have been addressed.